# A Chimeric IL-15/IL-15Rα Molecule Expressed on NFκB-Activated Dendritic Cells Supports Their Capability to Activate Natural Killer Cells [note 1]

**DOI:** 10.3390/ijms221910227

**Published:** 2021-09-23

**Authors:** Naomi C. Bosch, Lena-Marie Martin, Caroline J. Voskens, Carola Berking, Barbara Seliger, Gerold Schuler, Niels Schaft, Jan Dörrie

**Affiliations:** 1Institute of Medical Immunology, Martin-Luther University Halle-Wittenberg, 06112 Halle (Saale), Germany; naomi.c.bosch@gmail.com (N.C.B.); barbara.seliger@uk-halle.de (B.S.); 2Department of Dermatology, Universitätsklinikum Erlangen, Friedrich-Alexander University Erlangen-Nürnberg, 91054 Erlangen, Germany; l.m.martin@gmx.net (L.-M.M.); Caroline.Bosch-Voskens@uk-erlangen.de (C.J.V.); Carola.Berking@uk-erlangen.de (C.B.); gerold.schuler@uk-erlangen.de (G.S.); Niels.Schaft@uk-erlangen.de (N.S.); 3Comprehensive Cancer Center Erlangen–EMN, NCT WERA, 91054 Erlangen, Germany; 4Deutsches Zentrum Immuntherapie (DZI), 91054 Erlangen, Germany; 5Fraunhofer Institute for Cell Therapy and Immunology (IZI), 04103 Leipzig, Germany

**Keywords:** adoptive cellular immunotherapy, IL-15, natural killer cell, dendritic cell, NF-κB

## Abstract

Natural killer (NK) cells, members of the innate immune system, play an important role in the rejection of HLA class I negative tumor cells. Hence, a therapeutic vaccine, which can activate NK cells in addition to cells of the adaptive immune system might induce a more comprehensive cellular response, which could lead to increased tumor elimination. Dendritic cells (DCs) are capable of activating and expanding NK cells, especially when the NFκB pathway is activated in the DCs thereby leading to the secretion of the cytokine IL-12. Another prominent NK cell activator is IL-15, which can be bound by the IL-15 receptor alpha-chain (IL-15Rα) to be transpresented to the NK cells. However, monocyte-derived DCs do neither secrete IL-15, nor express the IL-15Rα. Hence, we designed a chimeric protein consisting of IL-15 and the IL-15Rα. Upon mRNA electroporation, the fusion protein was detectable on the surface of the DCs, and increased the potential of NFκB-activated, IL-12-producing DC to activate NK cells in an autologous cell culture system with ex vivo-generated cells from healthy donors. These data show that a chimeric IL-15/IL-15Rα molecule can be expressed by monocyte-derived DCs, is trafficked to the cell surface, and is functional regarding the activation of NK cells. These data represent an initial proof-of-concept for an additional possibility of further improving cellular DC-based immunotherapies of cancer.

## 1. Introduction

In the fight against cancer, therapeutic vaccination with dendritic cells (DCs) shows great potential. When patients’ DCs are loaded with tumor antigens, these are able to activate tumor-specific CD8^+^ cytotoxic T cells (CTLs), which then may eradicate the tumor [1]. Studies have shown that an additional innate immune response is highly beneficial when fighting the malignant tissue [2,3,4]. The standard protocol for cancer vaccination generates DCs from monocytes [5,6], which are subsequently matured using the standard cytokine cocktail consisting of TNFα, prostaglandin E2 (PGE_2_), IL-1β, and IL-6 [7]. However, this procedure alone is not sufficient to achieve the desired anti-tumor effect [1] for various reasons, which are still under debate. These include that DCs, used as a vaccine, do not reach the immunogenicity of a pathogen-driven inflammation, are often limited in IL-12 production, and do not properly interact with the innate immune system [8,9]. Additional activation of the NFκB pathway through electroporation with RNA encoding constitutively active (ca)IKKβ after maturation may give DCs the missing features [10,11]. Our group could already show that electroporation with caIKKβ leads to an upregulation of several activation markers, such as CD70, CD80 and CD40 and triggers the DCs’ ability to secrete bioactive IL-12 [10], a characteristic that merely cytokine cocktail-matured DCs lacked, but which is one of the most essential cytokines for the activation of the adaptive [12] and innate [13] immune response, especially against tumors. We found that NFκB-activated DCs displayed an enhanced ability to induce CTLs with a high lytic capacity and a memory-like phenotype [10] and to activate NK cells with the ability to lyse tumor cells [14]. Another pivotal cytokine for the activation of NK cells is IL-15 [15]. IL-15 itself is found in different functional forms either soluble or complexed with the IL-15 receptor α chain (IL-15Rα). This complex can provide IL-15 signaling *in trans* to adjacent cells [16].

The intracellular trafficking and processing of IL-15 is complex. Two premature isoforms are translated, which contain different signaling peptides, but are otherwise similar in their protein sequence. Nevertheless, processing and secretion of the isoforms follow different paths, which are individually regulated. Nevertheless, both require the presence of the IL-15Rα to reach the cell surface [17,18]. Hence, the secretion of IL-15 does not directly correlate with its transcription. 

Van den Bergh and his group showed that transfection with a combination of mRNA encoding IL-15Rα and mRNA encoding IL-15 resulted in a significantly increased NK-cell activation when compared to IL-15 alone [19]. Furthermore, the same stimulatory effect was observed when T cells were stimulated with either soluble IL-15/IL-15Rα complexes or with membrane bound IL-15/IL-15Rα presented on APCs [20]. Stoklasek et al. observed a robust proliferation of memory CD8^+^ T cells, NK cells and T cells in mice treated with soluble IL-15/IL-15Rα complexes [21]. On the other hand, systemic application of IL-15 in therapeutic applications is limited due to its high toxicity [22]. 

In continuation of the work of Van den Bergh et al. [19], we created a chimeric protein consisting of IL-15 and the IL-15Rα chain (chIL15), specially designed for efficient surface expression on DCs, which are intended for clinical application. The use of one chimeric protein instead of the two original proteins would simplify the DC-transfection process and the covalent link prevents the complex from dissociation. As initial proof-of-concept, we examine here whether expression of such a chimeric construct is feasible on DCs and if it could boost the potential of IL-12-producing DCs to activate NK cells. 

## 2. Results

We recently showed that cytokine cocktail-matured DCs (cmDC), electroporated with caIKKβ-encoding RNA, in order to activate the NFκB-pathway, were able to activate NK cells effectively [14], endowing them with the ability to secrete IFNγ and a lytic capacity towards target cells. Another relevant cytokine for the activation of NK cells is IL-15 [23], especially when IL-15 was trans-presented with the IL-15Rα [19]. We, therefore, designed a chimeric IL-15/IL-15Rα (chIL15) surface molecule to be transfected into DCs in addition to caIKKβ-RNA, and tested, whether this led to an enhanced NK-cell activation. 

### 2.1. chIL15 Is Expressed on chIL15-mRNA-electroporated DCs 

As it has been shown that the complex formation of IL-15 with its receptor is important for its efficacy [19,20], we designed a fusion protein of IL-15 and the IL-15Rα genetically fused via a flexible linker. Initially, we used a fusion protein consisting of the full-length IL-15 transcript 1 with its own signal peptide sequence linked to the mature IL-15Rα (chIL15old) (Figure 1A). After transfection of chIL15old into mDCs however, little IL-15Rα was detectable by surface marker staining with an IL-15Rα-specific antibody (Figure 1B, red line) and Western blot (Appendix A). We, therefore, postulated that the endogenous signal sequences are not suitable to mediate sufficient expression, processing, or trafficking [24] in DCs and therefore the chimeric protein was not properly expressed and shuttled to the cell surface. To overcome this we replaced the complete IL-15 signal peptide including the pro-peptide with the CD25 signal peptide (Figure 1A). CD25 was chosen for the following reasons: it is a surface molecule and not a soluble factor, it is a close homolog of the IL-15Rα, and it is efficiently expressed on the surface of mature DCs. To investigate whether this new chimeric IL-15/IL-15Rα (chIL15) leads to better surface expression, the protein was detected by surface staining with an IL-15Rα-specific antibody. Therefore, moDCs were matured with the standard cytokine cocktail and transfected with chIL15, or, as a control, were electroporated without adding any mRNA (mock). The chIL15 surface expression was determined via flow cytometry 2, 4, 6, and 24 h after electroporation. After 24 h over 50% of the cells stained positive (Figure 1B, blue line), and already after 2 h, a visible chIL15 expression was detected with an increase over 4 and 6 h until 24 h after electroporation (Figure 1C). The presence and expression of chIL15 could also be verified by Western blot analysis (Appendix A). We were, therefore, able to express the fusion protein efficiently on the DC’s surface. 

### 2.2. Stimulation with DCs Transfected with a Combination of caIKKβ and chIL15 Leads to an Enhanced Activation of NK Cells

To analyze whether the surface expression of chIL15 can increase the ability of caIKKβ-transfected cmDCs to activate NK cells, cmDCs were either electroporated with chIL15 RNA or caIKKβ RNA or with the combination of both. DCs electroporated without RNA (mock) were used as negative controls. The electroporated DCs were co-cultured for 48 h with autologous peripheral blood cells (PBMCs) at a ratio of 1:10 and, as additional controls, each cell type was cultured alone. 

To measure NK-cell activation, the cells were stained for CD3, CD56 and the well-established NK-cell activation markers CD69, CD25, and CD54 [25,26]. Cells were analyzed by flow cytometry and NK cells were identified within the mixed population of PBMCs by gating on CD3^–^/CD56^+^ cells, as this population represents the majority of *bona fide* NK cells [27] (see Appendix A for details). The mean fluorescence intensity (MFI) of each activation marker was calculated relative to the MFI of PBMCs alone. As we could already show previously [14], all three activation markers on NK cells were upregulated between 2- to 4-fold when PBMCs were co-cultured with caIKKβ-transfected DCs, reaching significance with regard to CD69 and CD25 (Figure 2A,B). In co-cultures, in which the DCs were co-transfected with chIL15 and caIKKβ, the expression of all three NK-cell activation markers was even higher (significant for CD69 and CD25), rising up to almost 5-fold (Figure 2A,B). Comparing co-cultures with mock and chIL15 DCs only, a slight increase in each activation marker was observed, which was statistically not significant. These data indicate that the activation of the main population of NK cells (CD3^–^/CD56^+^) by caIKKβ-transfected DCs can be further improved by additional IL-15 signaling.

As NK cells secrete large quantities of IFNγ upon activation [2], the concentration of this cytokine was quantified in the supernatants of the co-cultures. As expected, the secretion of IFNγ was significantly increased when PBMCs were co-cultured with caIKKβ-transfected DCs (Figure 2C). Co-cultures with DCs transfected with both caIKKβ and chIL15 secreted almost twice as much IFNγ (Figure 2C). DCs cultured alone and transfected with caIKKβ alone and those that were transfected with chIL15 and caIKKβ, also secreted lower quantities of IFNγ, while DCs electroporated only with chIL15-RNA lacked IFNγ secretion (Figure 2C). 

We previously demonstrated that both isolated NK cells and PBMCs co-cultured with caIKKβ-transfected DCs for one week had a superior lytic capacity against HLA-negative targets [14]. Since cytotoxicity is the most desirable property for an anti-tumor effect, we examined whether the co-electroporation of chIL15-RNA together with the caIKKβ-RNA would further increase the lytic capacity of the cells that had been co-cultured with such DCs against the classical HLA-negative NK-target cell line K562. This was tested in a standard ^51^chromium release assay with target-to-effector ratios of 1:6, 1:2, and 1:0.6. Cells from co-cultures with DCs electroporated with both chIL15- and caIKKβ-RNA were able to significantly lyse K562 cells in all target-to-effector ratios, and cells from co-cultures with DCs, electroporated with caIKKβ mRNA alone only marginally missed the formal *p*-value threshold (Figure 2D). Again, we observed an increase of lytic capacity against K562 cells when PBMCs were co-cultured with DCs that were electroporated with both chIL15- and caIKKβ-RNA compared to co-cultures with only caIKKβ-DCs, reaching the highest lytic capacity of 35% at a target-to-effector cell ratio of 1:6 (Figure 2D). The lytic capacity was even twice as high in co-cultures with chIL15- and caIKKβ-transfected DCs at the target-to-effector ratios of 1:2 and 1:0.6, reaching significance at a ratio of 1:0.6 (Figure 2D). As a control, PBMCs were cultured alone, but no difference to PBMCs co-cultured with mock-electroporated DCs was found (data not shown).

Taken together, the activation of NK cells by caIKKβ-transfected DCs can be further improved by additional IL-15 signaling resulting in an even higher upregulation of the activation markers CD69, CD25, and CD54 (Figure 2A,B), an enhanced secretion of IFNγ (Figure 2C), and a higher lytic capacity towards HLA-negative target cells (Figure 2D), albeit other cells in the co-culture may contribute to the latter two observations.

## 3. Discussion

Activating the NFκB pathway through electroporation with caIKKβ-RNA makes DCs ideal candidates for tumor vaccination, as they unite several beneficial features. Transfection with caIKKβ leads to a higher activation state of DCs providing them with the ability to secrete IL-12. This leads to an advanced stimulation capacity towards both CTLs and NK cells, which in turn gain a superior lytic capacity [10,14]. Currently, the efficacy of these DCs is evaluated in a phase I clinical trial (NCT04335890). Besides IL-12, as a crucial cytokine for the activation of effector cells, also IL-15 plays a pivotal role in the activation of both the adaptive and the innate immune responses [15,24,28,29]. Therefore, it represents an interesting candidate for improving immunotherapeutic approaches [30]. However, NFκB activation does not induce IL-15 secretion by mature monocyte-derived (mo)DCs [10] and merely adding soluble IL-15 has major disadvantages, as IL-15 has a very short half-life [21] and shows high transient toxicity in high plasma concentrations [22]. These negative features can be circumvented when IL-15 is bound to the IL-15Rα. Trans-presentation of IL-15 bound to the IL-15Rα shows high potential for the activation of the adaptive and the innate immune response. Treatment with nanoparticles trans-presenting IL-15 enhanced CD8^+^ T-cell potency [31]. Unfortunately, moDCs do not endogenously express the IL-15Rα but nevertheless showed enhanced activation of NK cells after being equipped with both IL-15 and the IL-15Rα by mRNA electroporation [19]. The fusion of IL-15 to the IL-15Rα might therefore further improve the activation of effector cells through DC vaccination. 

We observed that chIL15 alone had only a marginal effect on the activation status of NK cells, probably because IL-12 is critically required for NK activation and standard moDC, matured with the standard cytokine cocktail do not produce this cytokine. When, in contrast, transfected with chIL15 and caIKKβ-RNA, which facilitates IL-12-production, DCs were significantly better in NK cell activation. In a previous paper, we could already show that vast amounts of IFNγ that were secreted in PBMC and DC co-cultures were secreted by NK cells and not other bystander cells [14]. However, the results presented here, only provide an initial proof-of-principle and a deeper characterization of these DCs is necessary to address whether they maintain their T-cell stimulatory capacity, or are by any other means impaired in their immunogenic function. Also, the resulting NK cells should further be characterized for their activity against different tumor cells and their effector function in general. 

The use of a single chimeric construct for this purpose is advantageous for several reasons. Under good manufacturing practice (GMP) conditions, which are prerequisites for clinical application of cellular products, the production and transfection of one RNA vs. two separate ones is easier, cheaper and bears less variability. The covalent link prevents any dissociation of the IL-15 from the receptor α chain thus prolonging the activity and avoiding any free IL-15 that might act systemically. A suspected side effect of IL-15 is the induction of autoimmunity [32,33], and Sato et al. postulate that this effect requires *cis* presentation of IL-15 [33], and as chIL15 only presents IL-15 *in trans*, this side effect would be prevented. 

As mRNA electroporation is a well-established technique used in clinical trials on cancer immunotherapy [34], taking this approach from the bench to the bedside seems a reasonable task. DCs electroporated with caIKK-RNA are currently tested in a phase I clinical trial on uveal melanoma patients (NCT04335890). If these DCs prove safe and efficient, DCs additionally electroporated with the chIL15 could be clinically tested, provided that a reasonable risk assessment is performed and the regulatory authorities consider the risk:benefit ratio adequate. The intended activation of NK cells, however, would probably have the best impact in the treatment of tumors with impaired HLA expression, which renders the malignant cells invisible for CD8^+^ T-cell responses but increases their sensitivity to NK cells. Uveal melanoma, in contrast, usually shows good HLA class I expression [35]. Hence, tumor entities, which typically use HLA loss as a means of immune escape, would represent better targets. Cervical cancer, colorectal cancer, gastric cancer, and esophageal squamous cell carcinoma regularly display loss of HLA class I and this loss is often associated with progression or escape [36]. But for most other tumors, HLA-loss or down-regulation has been described to occur with different incidences. The possibility to generate DCs with an increased capacity to activate NK cells which in turn could block this immune escape route for the tumor provides a possibility to further individualize cancer immunotherapy and to specifically respond to tumor immune evasion mechanisms.

Taken together, our experimental data confirm that forced trans-presentation of IL-15 supports the activation of NK cells by DCs. Combining this with caIKKβ transfection into DCs extends their immune-stimulatory function, as it adds a missing piece to the system, overcoming the inability of such DCs to produce endogenous IL-15. This, on the one hand, confirms the importance of IL-15 trans-presentation in NK cell activation, and on the other hand, may represent a possibility to improve designer DC vaccines further. 

## 4. Materials and Methods

### 4.1. Acquisition of Primary Cells

Peripheral blood mononuclear cells (PBMCs) were isolated from 100 to 360 mL blood, taken from healthy donors after informed consent and approval by the institutional review board (Ethikkommission of the Friedrich-Alexander University Erlangen-Nürnberg, Ref. no. 4158), by density centrifugation with Lymphoprep (Axis-Shield PoC AS, Oslo, Norway) as described previously [37]. For the generation of moDCs, monocytes, were first separated from the non-adherent fraction (NAF) by plastic adherence, to be differentiated into immature DCs (iDCs) over the course of 6 days in DC medium, consisting of RPMI 1640 (Lonza, Verviers, Belgium) supplemented with 1% non-autologous human plasma (Sigma-Aldrich, St. Louis, MO, USA), 2 mM L-glutamine (Lonza), and 20 mg/l gentamycin (Lonza). Fresh DC medium with GM-CSF (800 IU/mL; Miltenyi Biotec, Bergisch Gladbach, Germany) and IL-4 (250 IU/mL; Miltenyi Biotec) was added on days 1, 3, and 5. On day 6, DCs were matured (mDC) with the standard cytokine cocktail consisting of 200 IU/mL IL-1*β* (CellGenix, Freiburg, Germany), 1000 IU/mL IL-6 (Miltenyi Biotec), 10 ng/mL TNF*α* (Beromun, Boehringer Ingelheim Pharma, Ingelheim am Rhein, Germany), and 1 μg/mL PGE_2_ (Pfizer, Zurich, Switzerland), as described in detail previously [14]. After 24 h of maturation, DCs were used for electroporation. Cells were incubated at 37 °C with 5% CO_2_.

### 4.2. RNA Constructs and Electroporation of DCs

For in vitro transcription of mRNA, the mMESSAGE mMACHINE^™^ T7 ULTRA Transcription Kit (Life Technologies, Carlsbad, CA, USA) was used and mRNA was purified with an RNeasy Kit (Qiagen, Hilden, Germany) according to the manufacturers’ instructions. To activate the classical NFκB pathway in DCs, RNA encoding a constitutively active mutant of IKKβ (caIKKβ) was used [10]. A DNA sequence encoding the original fusion construct consisting of the full-length IL-15 variant 1 (NCBI Reference Sequence: NP_000576.1) linked to the IL-15Rα (AA 31-267, NCBI Reference Sequence: NP_002180.1) with a flexible linker (SGGGSGGGGSGGGGSGGGGSGGGSLQ) was ordered from GeneART^®^ (Thermo Fisher Scientific, Waltham, MA, USA) and cloned into the pGEM4Z64A RNA production vector [37]. The sequence encoding the first 48 AA of the IL-15 was replaced with a sequence encoding the first 21 AA of CD25 (NCBI Reference Sequence: NP_000408.1) using two annealed complementary oligonucleotides (Eurofins Genomics, Ebersberg, Germany). The detailed sequences are provided as Appendix A.

DCs were electroporated in a total volume of 100 µL, with a maximum of 6 × 10^6^ DCs, using 30 µg of each mRNA, with a square-wave pulse and 1250 V/cm for 1 ms, as described in detail [38]. As a control, cells were electroporated under identical conditions but without any RNA (mock). 

### 4.3. moDCs and PBMCs Co-Cultures

After transfection, DCs were rested for 2–4 h, then 2 × 10^5^ DCs/mL and 2 × 10^6^ fresh autologous PBMCs/mL were co-cultured, gaining a ratio of 1:10. A maximum of 2 × 10^5^ DCs and 2 × 10^6^ PBMCs, and a minimum of 4 × 10^4^ DCs and 4 × 10^5^ PBMCs were used. PBMCs cultured alone served as control. Cells were seeded into 24- or 48-well plates, depending on cell numbers and incubated in MLPC medium consisting of RPMI 1640 (Lonza), 10% non-autologous human serum (Sigma-Aldrich), 2 mM L-glutamine (Lonza), 20 mg/L gentamycin (Lonza), 10 mM HEPES (PAA Laboratories, GE Healthcare Life Sciences, Pasching/Linz, Austria), 1 mM sodium pyruvate (Lonza), and 1% non-essential amino acid (100×; Lonza). Cells and supernatants were harvested after 48 h. To generate cells for cytotoxicity assays, co-cultures were extended to 1 week and during this period cells were split and fresh MLPC medium was added as necessary. 

### 4.4. Analysis of Marker Expression on the Cell Surface

Cells were harvested either 2, 4, 6, and 24 h after electroporation or after 48 h of co-culture. The expression of surface markers was determined via flow cytometry using anti-CD215-PE (anti-human IL-15Rα, clone JM7A4) (Biolegend, San Diego, CA, USA) and IgG2b-PE isotype control, anti-CD56-FITC, anti-CD3-APC-Cy7 or anti-CD3-V500, anti-CD69-PE, anti-CD25-BV421 or anti-CD25-PE, and anti-CD54-APC or anti-CD54-PE (all from BD Biosciences, Heidelberg, Germany) as described [39]. A FACS Canto II flow cytometer (BD Biosciences) and FACSDiva software [40] were used to measure immunofluorescence and acquire data, which was evaluated with FCS Express software [41].

### 4.5. Measurement of Cytokine Secretion

The supernatants of the co-cultures were sampled after 48 h. The concentration of IFNγ was determined using the Human Th1/Th2 Cytometric Bead Array Kit II (BD Biosciences) following the manufacturer’s instructions. A FACS Canto II flow cytometer and BD FACSDiva software [40] were used to measure immunofluorescence and acquire data, which was evaluated with FCS Express software [41]. 

### 4.6. Cytotoxicity Assay

Co-cultures were harvested after 1 week of co-incubation. Then the cytolytic capacity was determined in a standard 4–6 h ^51^chromium release assay as described in detail before [42]. In short, the target cell line K562 was labeled with 100 μCi of Na_2_^51^CrO_4_/10^6^ cells (PerkinElmer, Waltham, MA, USA), washed and subsequently cultured in 96-well plates (Thermo Fisher Scientific) at 1000 cells/well. Effector cells harvested from the co-cultures were added at E:T ratios of 6:1, 2:1 and 0.6:1 (i.e., 6000, 2000, and 600 cells per well, respectively). The supernatant was taken after 4–6 h and the release of chromium was measured via a Wallac 1450 MicroBeta plus Scintillation Counter (Wallac, Turku, Finland). The following formula was applied to calculate the percentage of lysis: (measured release − background release)/(maximum release − background release) × 100%.

### 4.7. Statistical Analysis

GraphPad Prism [43] was employed to create graphs and statistical analysis. To determine statistical significance and *p* values, a paired one-way ANOVA, and subsequent multiple comparison analyses using the Tukey test were performed, assuming Gaussian distribution, based on our experience in similar experiments. To analyze the expression kinetic of chIL15, a two-way ANOVA was performed.

## Figures and Tables

**Figure 1 ijms-22-10227-f001:**
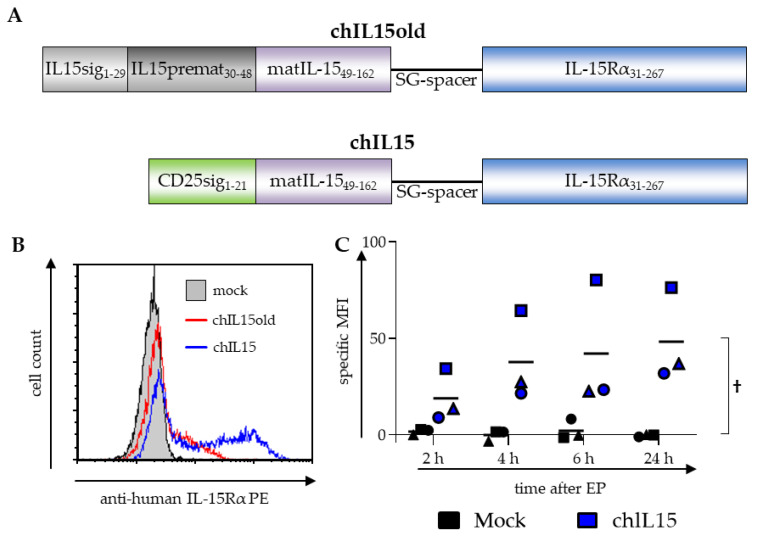
Composition and expression of the chimeric IL15 constructs. (**A**) Originally, the protein sequence of the complete IL-15 transcript variant 1 was fused to that of the IL-15Rα via a flexible SG-linker (SG_3_(SG_4_)_3_SG_3_S). This construct is referred to as chIL15old. To improve the construct, the first 48 AA of the IL-15 transcript variant 1 that constitute the signal peptide and the pro-peptide were replaced with the signal sequence of CD25. The signal sequence of the IL-15Rα was removed so that the linker was directly fused to the first sushi-domain. This altered construct is referred to as chIL15. (**B**) To measure the expression of the chIL15old and the chIL15 construct on the surface of DCs, cytokine matured DCs were electroporated with mRNA encoding the chIL15old (red line) or chIL15 (blue line), or as a negative control were mock electroporated (grey histogram). The cells were stained with an IL-15Rα-specific antibody and analyzed by flow cytometry. A histogram of DCs electroporated with chIL15old, chIL15, or without RNA (mock) and stained for IL-15Rα 24 h after electroporation is shown (*n* = 3 for chIL15, *n* = 1 for chIL15old). (**C**) The expression kinetics of chIL15 was determined 2, 4, 6, and 24 h after electroporation. The specific mean fluorescence intensity (MFI) was calculated by subtracting the isotype control MFI from the MFI measured with the IL-15Rα-specific antibody (different symbols represent the three individual experiments). The interaction *p*-value between the mock and the chIL15 condition was calculated by two-way ANOVA, † *p* ≤ 0.05.

**Figure 2 ijms-22-10227-f002:**
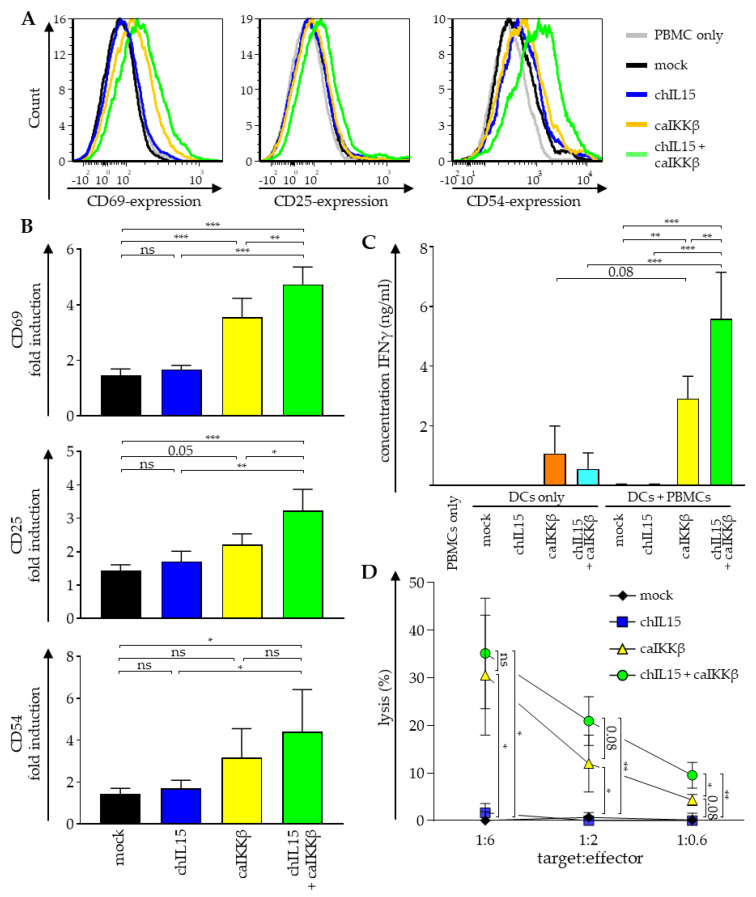
Transfection of DCs with caIKKβ combined with chIL15 leads to an improved NK-cell activation. DCs matured with the standard cytokine cocktail were electroporated with RNA encoding caIKKβ and chIL15 either alone or in combination. As a control, DCs were mock electroporated. Transfected DCs were co-cultured with autologous peripheral blood mononuclear cells (PBMCs) 2–4 h after electroporation at a ratio of 1:10 (final concentrations 2 × 10^5^ DCs/mL and 2 × 10^6^ PBMCs/mL). As controls, transfected DCs and PBMCs were cultured alone. Cells were harvested and supernatant was sampled after 48 h of co-culture. (**A**) The expression of surface markers CD69, CD25, and CD54 was determined via flow cytometry on cells negative for CD3 and positive for CD56 (using the gating strategy shown in Appendix A). A representative histogram for each activation marker out of three independent experiments is shown. (**B**) The depicted values show the fold upregulation of each surface marker, calculated relative to the mean fluorescence intensity (MFI) of PBMCs cultured in absence of DCs. The average fold induction of three different donors with SD is shown. *p* values were calculated with a one-way ANOVA and subsequent Tukey test, *** *p* ≤ 0.001, ** *p* ≤ 0.01, * *p* ≤ 0.05, numbers indicate *p* values of 0.05 < *p* ≤ 0.1. (**C**) The secretion of IFNγ was measured in the supernatant by Cytometric Bead Array. Average cytokine concentrations with SD are shown from three independent donors. *p* values were calculated with a one-way ANOVA and subsequent Tukey test, *** *p* ≤ 0.001, ** *p* ≤ 0.01, numbers indicate *p* values of 0.05 < *p* ≤ 0.1. (**D**) The cytolytic capacity of stimulated cells was determined in a ^51^chromium release assay. The K562 cell line was used as a target at the indicated target ratios. Average values with SD of three independent donors, each analyzed in triplicates, are shown. *p* values were calculated with a one-way ANOVA and subsequent Tukey test, ** *p* ≤ 0.01, * *p* ≤ 0.05, numbers indicate *p* values of 0.05 < *p* ≤ 0.1. The respective brackets refer to the mock-transfected co-cultures.

## Data Availability

All original data are available from the corresponding author on request.

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
