# Peer review of "A Chimeric IL-15/IL-15Rα Molecule Expressed on NFκB-Activated Dendritic Cells Supports Their Capability to Activate Natural Killer Cells"

_ijms, 2021, doi:10.3390/ijms221910227_

Round 1

Reviewer 1 Report

The manuscript describes a new methodology on natural killer (NK) cells activation through an electroporation of mRNA coding a chimeric protein with interleukin (IL)-15 and IL-15 receptor (IL15R) alpha. Although the findings are interesting, some points are poor enough to claim a scientific originality and significance. Major my concerns are listed below:

  1. In the Reference 16, Stoklasek et al. reported the IL-15/IL-15Ralpha enhanced an activation of NK cells as well as CD8-positive T cell. I don’t understand the originality of this paper. The authors should clarify the originality of the manuscript in Abstract, Introduction and Conclusion.
  2. In Figure 2A the authors didn’t seem to distinguish NK cells and other cells. In the Reference 18 that the authors cited, CD3-negative fraction in PBMC was regarded as NK cells. As far as I know, CD3-negative and CD56, CD94 or NKp46-positive cells are recognized as NK cells. The authors should specify NK cell population in PBMC using these marker? If no, the authors should explain the reason.
  3. The reason why CD25 signal sequence was used for secretion is unclear. The authors should indicate the rationale for CD25 signal sequence.

Author Response

The manuscript describes a new methodology on natural killer (NK) cells activation through an electroporation of mRNA coding a chimeric protein with interleukin (IL)-15 and IL-15 receptor (IL15R) alpha. Although the findings are interesting, some points are poor enough to claim a scientific originality and significance. Major my concerns are listed below:

Answer: Thank you very much for taking the time and effort to review our manuscript. We have addressed your concerns below and modified the manuscript accordingly.

1. In the Reference 16, Stoklasek et al. reported the IL-15/IL-15Ralpha enhanced an activation of NK cells as well as CD8-positive T cell. I don’t understand the originality of this paper. The authors should clarify the originality of the manuscript in Abstract, Introduction and Conclusion.

Answer: This is a valid point as we had only discussed the advances of our approach in the discussion section. Stoklasek et al. only report the use of a soluble IL-15/IL-15Ralpha fusion protein, which is of very limited use in therapeutic application due to its toxicity. Our intent, in contrast, was to further develop DC-based immunotherapy, which requires a DC-specific surface expression; at best, using a GMP-compliant methodology. We have modified the abstract (line 28-33), the introduction (line 76-85), and the discussion (line 263, 264), to better clarify out intention.

2. In Figure 2A the authors didn’t seem to distinguish NK cells and other cells. In the Reference 18 that the authors cited, CD3-negative fraction in PBMC was regarded as NK cells. As far as I know, CD3-negative and CD56, CD94 or NKp46-positive cells are recognized as NK cells. The authors should specify NK cell population in PBMC using these marker? If no, the authors should explain the reason.

Answer: The cells, whose activation markers are shown in Fig. 2A were gated to be CD3-negative and CD56-positive. Reference 18 (now 25) also performs double staining for both CD3 and CD56. This is the traditional definition of NK cells, although other markers have been described that further define NK subpopulations, including CD56-dim NK-cells. Even CD56-negative NK-cells were found, but predominantly not in healthy donors. Nevertheless, our intention was not to cover all different NK subtypes, but to show that a population, that consists of bona-fide NK-cells is activated. We modified the result section (line 149-151) and the figure legend (line 198-199) accordingly.

3.The reason why CD25 signal sequence was used for secretion is unclear. The authors should indicate the rationale for CD25 signal sequence.

Answer: Thank you for pointing that out. We were indeed a bit short on our rational: i) CD25 is a surface protein, and not a soluble factor, and we want surface expression, ii) CD25 is the closest homologue of the IL-15Ralpha, and iii) CD25 is efficiently expressed in the type of DCs, we are using. We included this rational in the result section (line 106-110).

Reviewer 2 Report

The manuscript by Bosch et al. describes the generation of DCs expressing a chimeric IL15/IL15Ra protein. These cells are used in cocultures to activate NK cells that exhibit superior cytolytic activities against MHCI-negative tumour cells. The ideas in the paper are novel and will appeal to a wide range of researchers, offering a new strategy for generation of more effective DC vaccines. However, there are a few technical issues with data presentation, and many findings in the paper are referred to as ‘data not shown’. The authors need to present all of their data, even the negative findings, as these help shape the context of the experiments that are presented. Furthermore, I am not convinced that the authors have produced adequate data to support their conclusion that the DCs are activating NK cells. I have raised some queries and suggested some experiments in my comments to address this.

Comments:

Line 19: Should say “adaptive immune system” rather than “adaptive immunity”        

Line 42: Please state why this procedure does not induce the desired antitumour effect – is it due to a lack of co-stimulatory molecule upregulation?

Line 56: Do the two premature isoforms of IL15 generate the soluble and complexed forms of IL15, respectively? Please make this clear. Perhaps move the final sentence of the previous paragraph into this paragraph.

Line 52-66: Is anything known about the functional role of soluble and complexed IL15 in a ‘natural’ immune response? Are they released at different times, thus promoting different levels of NK responsiveness during the progression of the immune response? Do they act on different cells, with differing roles? Suggest commenting on this.

Line 87/Fig1: Please show this data and it’s associated methods, in fig1 if possible, or at least as supplementary data. It is often just as important to show negative data as positive data, and there is plenty of space in the paper for this data to be included. If this data is included, it becomes clearer to the reader why the second construct was better. Also suggest including a diagram of the initial construct with the second construct in Fig 1a.

Line 90: please indicate that the sequence removed from IL15 included both the SigP and the pro-peptide

Line 95: It might be useful to define here what you mean by mock electroporation.

Line 100: As above, please show this data in fig 1.

Fig 1B: Please indicate statistical significance between mock and experiment, measured by ANOVA, on this plot. How many replicates is this? The authors could present their column data in both figures as column scatter graphs to make this clear.

Figure 1 legend: Please define the y-axis of Figure 1b (i.e. MFI) and indicate what the error bars represent. Suggest using standard deviation.

Fig 1c: Can the authors please explain why the flow plots show two peaks in the experimental sample? Is this due to incomplete transfection? Or is the surface expression of the fusion variable across transfected cells? Did the authors use a selection drug to select for successfully transfected DCs? If so please state this in the methods.

Line 117: Why was 48 hours chosen for NK activation? NK cells can become activated in culture within a few hours. Would you expect different results with shorter periods of activation?

Figure 2a: Could the authors please show some histogram plots to demonstrate the data presented in the tables? An overlay for each marker, with the four comparisons will be sufficient. Perhaps the authors could split figure 2 into two figures, with fig 2b and c becoming fig 3, to allow room for these plots.

Figure 2: Why was a t-test used for these comparisons? As you’ve got more than one comparison, and ANOVA with multiple comparisons should be used to determine statistical significance.

Figure 2 : Please use standard deviation rather than SEM. SEM only tells you about the accuracy of the mean. SD tells you about the variability of observations, which is more important for comparisons and statistical significance.

Line 156: Should say Figure 2B

Line 165: Can the authors please comment on why it was necessary to coculture the DCs and PBMCs for one week prior to cytotoxicity? As previously mentioned, I would have expected the NK cells to be activated much more quickly. Can the authors be sure that this killing is NK mediated? There are antigen-independent ‘virtual memory CD8+ T cells’ that can act as effectors and are highly responsive to IL15. I think if the authors believe this to be NK killing, it is necessary to provide some evidence of this given that mixed PBMCs were used. The author could FACS sort out the NK cells based on marker expression to repeat their killing assays. Alternatively, if this is not a possibility, could the authors do some fluorescent microscope work with labelled markers to show the NK cells killing the K562s? I am not convinced, based on timing, that this is NK killing that is being detected.

Fig 2c: Is this effect specific to MHCI-negative cancer cells? Did the authors try any other cancer cell lines? Generating this data would increase the depth of the findings and give the paper more impact by demonstrating that the activated NK cells are effective killers across multiple cancers.

Results: Have the authors considered trying the DCs in a mouse model of cancer to provide more evidence of the therapeutic potential of their generated cells?

Line 202: Results should be discussed, not restated. Do not need to refer to figures in discussion.

Line 205: These are results and do not belong in the discussion section. Why are the CD8 results not presented in the main paper or supplementary data? Negative results are just as important as positive ones – we cannot believe what you are saying unless the evidence is provided. CD8 T cells have not previously been discussed in the paper, so this statement comes as a surprise.

Line 204-211: Rather than restating the results, can the authors please discuss their results in the context of previous studies. Are there any other studies that have shown similar results? What literature exists on IL15 to help the authors explain their results?

Discussion: Could the authors please add some more discussion on the therapeutic potential of this DC vaccine at this stage. Where is it likely to be most effective? What still needs to be done to get it to trials?

Line 257: Please indicate that the fusion construct was also a RNA construct

Line 258-264: Please provide a Genbank or Ensembl ID for the IL15, IL15Ra and CD25 sequences used.

Line 268: Can the authors please confirm whether the mock DCs were generated with just the chIL15 RNA excluded? Or both chIL15 and caIKKb RNA excluded?

Methods general: please indicate how many cells were used in each assay.

Line 312: The authors guidelines have been left in the paper and should be removed.

General comment: There are some minor issues with grammar and sentence structure throughout the manuscript that could be addressed by the authors to improve readability.

Author Response

The manuscript by Bosch et al. describes the generation of DCs expressing a chimeric IL15/IL15Ra protein. These cells are used in cocultures to activate NK cells that exhibit superior cytolytic activities against MHCI-negative tumour cells. The ideas in the paper are novel and will appeal to a wide range of researchers, offering a new strategy for generation of more effective DC vaccines. However, there are a few technical issues with data presentation, and many findings in the paper are referred to as ‘data not shown’. The authors need to present all of their data, even the negative findings, as these help shape the context of the experiments that are presented. Furthermore, I am not convinced that the authors have produced adequate data to support their conclusion that the DCs are activating NK cells. I have raised some queries and suggested some experiments in my comments to address this.

Answer: Thank you very much for the extensive review and correction of our manuscript. This is really appreciated.

Comments:

Line 19: Should say “adaptive immune system” rather than “adaptive immunity”        

Answer: We have changed that. (line 19)

Line 42: Please state why this procedure does not induce the desired antitumour effect – is it due to a lack of co-stimulatory molecule upregulation?

Answer: We included more information about this in the introduction (line 46-48). Generally, it is assumed that the problem is rather a lack of cytokine-production (e.g. IL-12).

Line 56: Do the two premature isoforms of IL15 generate the soluble and complexed forms of IL15, respectively? Please make this clear. Perhaps move the final sentence of the previous paragraph into this paragraph.

Answer: Indeed it seems that IL-15 originating from both forms can only be shuttled to the cell surface as complex with the IL-15Ra, but the release of IL-15 derived from the shorter version seems to be regulated even more stringently. We have modified the text, and also mentioned the concept of trans-presentation earlier (line 65-66), we furthermore included 3 new references (16-18). On the other hand, since this is meant to be a short communication, we do not want to elaborate too much.

Line 52-66: Is anything known about the functional role of soluble and complexed IL15 in a ‘natural’ immune response? Are they released at different times, thus promoting different levels of NK responsiveness during the progression of the immune response? Do they act on different cells, with differing roles? Suggest commenting on this.

Answer: Please see response to last comment.

Line 87/Fig1: Please show this data and it’s associated methods, in fig1 if possible, or at least as supplementary data. It is often just as important to show negative data as positive data, and there is plenty of space in the paper for this data to be included. If this data is included, it becomes clearer to the reader why the second construct was better. Also suggest including a diagram of the initial construct with the second construct in Fig 1a.

Answer: The reviewer is absolutely right about negative results. We now included the western blots as supplementary data S1 and the FACS-data in figure 1C. We also added the design of the original construct in figure 1A. (Actually we had those data in a draft of the manuscript, but had removed them).

Line 90: please indicate that the sequence removed from IL15 included both the SigP and the pro-peptide

Answer: We now indicate that in the text (line 107) and the figure legend (line 127).

Line 95: It might be useful to define here what you mean by mock electroporation.

Answer: We now indicate that we meant electroporation without mRNA (line 114).

Line 100: As above, please show this data in fig 1.

Answer: Done

Fig 1B: Please indicate statistical significance between mock and experiment, measured by ANOVA, on this plot. How many replicates is this? The authors could present their column data in both figures as column scatter graphs to make this clear.

Answer: We apologize for not indicating this previously. The data represent 3 independent experiments. To show the individual values, we now use a scatter plot. A 2-way ANOVA test revealed that the mock condition and the chIL15 condition differ significantly (p = 0.015), although the individual time points are above the formal significance threshold of 0.05. (between 0.2 and 0.07). The figure and figure legend (line 137) were modified accordingly.

Figure 1 legend: Please define the y-axis of Figure 1b (i.e. MFI) and indicate what the error bars represent. Suggest using standard deviation.

Answer: See above, Fig 1B (now Figure 1C) was changed to a scatter dot plot. The legend now indicates how specific MFI was determined (line 134).

Fig 1c: Can the authors please explain why the flow plots show two peaks in the experimental sample? Is this due to incomplete transfection? Or is the surface expression of the fusion variable across transfected cells? Did the authors use a selection drug to select for successfully transfected DCs? If so please state this in the methods.

Answer: The successfully transfected cells were not selected or gated in any way. The reason that two peaks (a negative and a positive peak) are observed results from the fact that the transfection efficiency is not 100 % (in this case about 50%).

Line 117: Why was 48 hours chosen for NK activation? NK cells can become activated in culture within a few hours. Would you expect different results with shorter periods of activation?

Answer: We also analyzed cells after 24 h and after 1 week. The results after 24h were similar to 48h, however, there was a staining issue with one donor, so the data are just n=2. After 1 week, the expression of the activation markers is to variable for a meaningful statistical analysis.

Figure 2a: Could the authors please show some histogram plots to demonstrate the data presented in the tables? An overlay for each marker, with the four comparisons will be sufficient. Perhaps the authors could split figure 2 into two figures, with fig 2b and c becoming fig 3, to allow room for these plots.

Answer: We included a new panel in figure 2 which shows representative histogram overlays (new Figure 2A).

Figure 2: Why was a t-test used for these comparisons? As you’ve got more than one comparison, and ANOVA with multiple comparisons should be used to determine statistical significance.

Answer: The reviewer is absolutely right, an ANOVA is the better choice for multiple comparisons and also has more power. We have reanalyzed all data using a one-way ANOVA. This resulted in over-all better p-values but confirmed that chIL-15 plus caIKKb was superior to caIKK alone, while chIL15 alone had no significant effects. Figure 2, the figure legend (line 202, 205, 209), and the material and method section (line 370, 371) were modified accordingly.

Figure 2 : Please use standard deviation rather than SEM. SEM only tells you about the accuracy of the mean. SD tells you about the variability of observations, which is more important for comparisons and statistical significance.

Answer: SEM was changed to SD

Line 156: Should say Figure 2B

Answer: Thank you for spotting that.

Line 165: Can the authors please comment on why it was necessary to coculture the DCs and PBMCs for one week prior to cytotoxicity? As previously mentioned, I would have expected the NK cells to be activated much more quickly. Can the authors be sure that this killing is NK mediated? There are antigen-independent ‘virtual memory CD8+ T cells’ that can act as effectors and are highly responsive to IL15. I think if the authors believe this to be NK killing, it is necessary to provide some evidence of this given that mixed PBMCs were used. The author could FACS sort out the NK cells based on marker expression to repeat their killing assays. Alternatively, if this is not a possibility, could the authors do some fluorescent microscope work with labelled markers to show the NK cells killing the K562s? I am not convinced, based on timing, that this is NK killing that is being detected.

Answer: We chose this time-point because for some donors, the NK cells (i.e. the CD3-negative CD56-positive fraction) of the lymphocytes clearly expanded (from 15% to almost 40%). However, this effect was quite donor specific. Nevertheless, we chose this time point to allow for this expansion, as it also has been done in previous publications (Massa et al. 2013 JI https://doi.org/10.4049/jimmunol.1202024 and Bosch et al. 2019 Ther. Adv. Med. Oncol. https://doi.org/10.1177/1758835919891622). To meet the factually correct concerns of the reviewer, we have restated the results and now indicate that we are measuring lytic activity against the classical HLA-negative NK-target cell line K562, and mention that this activity is not necessarily exerted by NK cells (line 177, 178, 214, 215).

Fig 2c: Is this effect specific to MHCI-negative cancer cells? Did the authors try any other cancer cell lines? Generating this data would increase the depth of the findings and give the paper more impact by demonstrating that the activated NK cells are effective killers across multiple cancers.

Answer: Thank you for this reasonable suggestion, we will surely include this idea in the next steps on the way to clinical application. This short communication is intended to show just an initial proof-of-principle, therefor we used the best established NK-target here, but mention this as subsequent experiments in the discussion (line 253-255).

Results: Have the authors considered trying the DCs in a mouse model of cancer to provide more evidence of the therapeutic potential of their generated cells?

Answer: This actually opens a long and interesting discussion about murine cancer models for DC-vaccination. Actually, we think that the fine-tuning of the DC vaccine is not possible in mice. Therefore we would rather test such designer DC in a small phase I clinical trial, as we are currently doing with the caIKK-activated DCs (NCT04335890).

Line 202: Results should be discussed, not restated. Do not need to refer to figures in discussion.

Answer: We removed the whole paragraph, as it indeed just restates results and our rational to remove the signal sequence (line 237-240).

Line 205: These are results and do not belong in the discussion section. Why are the CD8 results not presented in the main paper or supplementary data? Negative results are just as important as positive ones – we cannot believe what you are saying unless the evidence is provided. CD8 T cells have not previously been discussed in the paper, so this statement comes as a surprise.

Answer: You are correct, we cannot discuss results we do not present. Therefor the statement was removed from the discussion (line 241-243).

Line 204-211: Rather than restating the results, can the authors please discuss their results in the context of previous studies. Are there any other studies that have shown similar results? What literature exists on IL15 to help the authors explain their results?

Answer: see above, the whole paragraph was replaced.

Discussion: Could the authors please add some more discussion on the therapeutic potential of this DC vaccine at this stage. Where is it likely to be most effective? What still needs to be done to get it to trials?

Answer: We have extended the discussion on three aspects. 1) What still needs to be done on the bench to further characterize the DCs and the resulting NK cells (line 251-255), 2) How the road to clinical application would look like (line 265-271), and 3) Which tumor entities may be most promising (line 275-280).

Line 257: Please indicate that the fusion construct was also a RNA construct

Answer: Sorry for being unclear, we clarified that: actually at this stage, it is still a DNA construct, from which the RNA is transcribed in vitro. We apologize for being unclear on that.

Line 258-264: Please provide a Genbank or Ensembl ID for the IL15, IL15Ra and CD25 sequences used.

Answer: We now included as supplementary information the complete sequences of both constructs and provided the NCBI Reference Sequence identifier for all transcripts (Line 316-318, 322).

Line 268: Can the authors please confirm whether the mock DCs were generated with just the chIL15 RNA excluded? Or both chIL15 and caIKKb RNA excluded?

Answer: Mock always refers to cells electroporated without any RNA. We have stated that now more clearly at several occasions throughout the manuscript.  

Methods general: please indicate how many cells were used in each assay.

Answer: Numbers and volumes are now all included throughout the material and method section.

Line 312: The authors guidelines have been left in the paper and should be removed.

Answer: Sorry, removed.

General comment: There are some minor issues with grammar and sentence structure throughout the manuscript that could be addressed by the authors to improve readability.

Answer: We went through the manuscript and corrected for readability.

Round 2

Reviewer 1 Report

The revised version manuscript that the authors prepared is consistent with my criteria.